# Vector-Quantized Prompt Learning for Paraphrase Generation

**Haotian Luo**[*]
Sichuan University
haotianluo2002@gmail.com

**Yixin Liu**[*]
Sichuan University
liuyixin22@stu.scu.edu.cn

**Peidong Liu**
Sichuan University
hugh@stu.scu.edu.cn

**Xianggen Liu**[†]
Sichuan University
liuxianggen@scu.edu.cn

## Abstract

Deep generative modeling of natural languages has achieved many successes, such as producing fluent sentences and translating from one language into another. However, the development of generative modeling techniques for paraphrase generation still lags behind largely due to the challenges in addressing the complex conflicts between expression diversity and semantic preservation. This paper proposes to generate diverse and high-quality paraphrases by exploiting the pre-trained models with instance-dependent prompts. To learn generalizable prompts, we assume that the number of abstract transforming patterns of paraphrase generation (governed by prompts) is finite and usually not large. Therefore, we present vector-quantized prompts as the cues to control the generation of pre-trained models. Extensive experiments demonstrate that the proposed method achieves new state-of-art results on three benchmark datasets, including Quora, Wikianswers, and MSCOCO. We will release all the code upon acceptance.

## 1 Introduction

Paraphrase generation aims to produce sentences that have different expressions but convey the same semantic meaning given a particular sentence. Paraphrasing is a common phenomenon that reflects the diversity of human languages, serving as an important research topic in natural language processing. It has broad applications such as in question answering (Mckeown, 1983) and information retrieval (Knight and Marcu, 2000). However, automatically generating accurate and different-appearing paraphrases is still very challenging, since it requires the abilities of both understanding and generation.

Conventional methods draw on rule-based systems (Mckeown, 1983; Barzilay and Lee, 2003; Zhao et al., 2009; Lin and Pantel, 2001) and statistical machine translation (Quirk et al., 2004; Zhao et al., 2008) to generate paraphrases. These methods are easy to interpret and analyze, but struggle to yield fluent and diverse sentences. Recently the accumulation of the paraphrase data provides an unprecedented opportunity to directly learn the paraphrasing transformations in an end-to-end manner (Vaswani et al., 2017). For instance, Wang et al. (2019) formulate paraphrasing as a supervised encoding-decoding problem and use stacked residual LSTM networks to generate paraphrases.

A good paraphrase is a sentence that shares similar semantics but has noticeable syntactical or lexical differences from the original one (Lin and Wan, 2021). To improve the diversity of generated sentences, (Gupta et al., 2018) introduce the variational auto-encoder (VAE) to perform paraphrase generation. (Li et al., 2018) propose multiple generators with different granularity levels to learn the mapping relationship between input and output respectively, and then combine them to complete the paraphrase generation task. But those generated paraphrases tend to only make trivial changes to original sentences, such as modifications of synonyms.

Further, Hosking and Lapata (2021) leverage autoencoder to encode the structure and semantics of the sentence separately, and generate paraphrases by perturbing the structure encoding. Liu et al. integrate the word editing and rule-based transformation operations into deep learning and achieve the previous SOTA performance in paraphrase generation (Liu et al., 2022, 2020). However, due to the limitation of scales of the paraphrasing datasets, neural networks tend to generate the paraphrases with local changes to the inputs rather than global modifications on sentence structures.

In this work, we aim to exploit the knowledge of the pre-trained language model to balance expression diversity and semantic preservation. There-

---

[*]These authors contributed equally to this work.
[†]Corresponding authors.

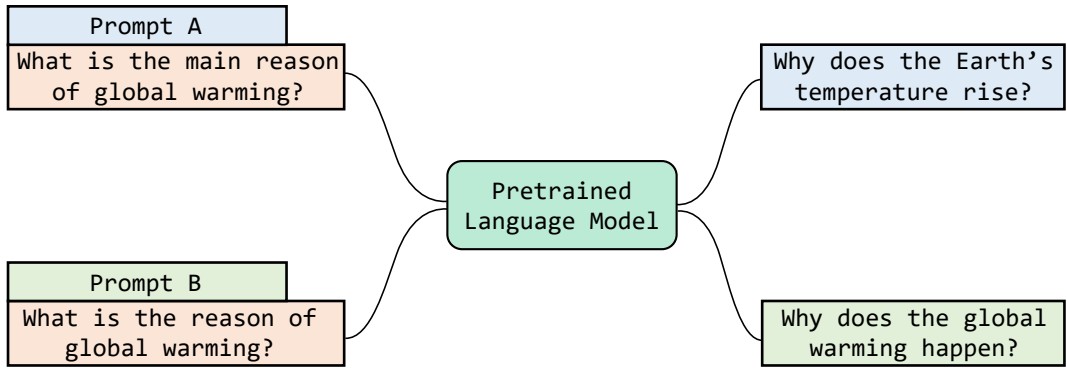

Figure 1: An example of the ideal prompt that induces the pre-trained language model to generate particular paraphrases. The proposed VQPrompt model aims to learn such prompts for each given sentence.

fore, inspired by (Bhardwaj et al., 2022) we propose a vector-quantized prompt learning framework, called VQPrompt, to generate diverse and high-quality paraphrases. In particular, VQPrompt comprises a prompt encoder and a pre-trained generative language model. The prompt encoder produces discrete prompts and the generative language model accepts both the prompts and the input sentence to generate the corresponding paraphrases. To make the vector-quantization work, we also introduce a K-means training strategy to dynamically update the codebook in the prompt encoder.

We evaluate the effectiveness of our model on four paraphrasing datasets, namely, Quora, Wikianswers, and MSCOCO. Experimental results show that VQPrompt achieves a new state-of-the-art paraphrasing performance in terms of both automatic metrics and human evaluation. In summary, our contributions are as follows:

- We propose vector-quantized prompt learning to adapt large pre-trained language models for paraphrase generation.

- We introduce a K-means training strategy to dynamically update the codebook in vector quantization (VQ), addressing the index collapse of VQ.

- The proposed method achieves new state-of-the-art performances in three benchmark datasets and presents modest interpretability.

## 2 Related Work

One of the characteristics of the paraphrase generation task is that there exist several general transformation rules. Therefore, rule-based methods have been used for paraphrase generation tasks

as early as the last century. Representative methods include dictionary-based and template-based methods. Dictionary-based methods look up synonyms in the dictionaries such as HowNet (Dong and Dong, 2003) or WordNet (Miller, 1995) to replace words in the original sentence, thereby generating corresponding paraphrases (Kauchak and Barzilay, 2006). The advantage of such a rule-based approach is that it is interpretable and controllable. Its shortcomings lie in the heavy workload of manually writing rules, and the generated sentences are not smooth enough.

With the accumulation of paraphrase corpus, researchers then start to model the paraphrase generation task as a single-language statistical translation process, thereby improving the fluency of generating paraphrase sentences (Quirk et al., 2004; Zhao et al., 2009). The statistical translation model learns the transition probability from the original sentence to the paraphrases from a large amount of training data. For example, Dolan et al. collected a large number of news texts from the Internet and built a paraphrase generation corpus, and then used statistical machine translation methods to generate paraphrase sentences (Dolan et al., 2004). However, statistical paraphrasing methods still require heavy feature engineering.

In recent years, deep neural network has become the mainstream paraphrase generation method due to its powerful fitting ability (Chowdhury et al., 2022; Hosking and Lapata, 2021). Similar to the statistical paraphrasing methods, the neural-based paraphrase generation method formulates the paraphrase generation as a single-language translation process, but adopts an encoding-decoding network structure and an end-to-end training method. The first deep paraphrasing method takes the long

short-term memory network LSTM (Hochreiter and Schmidhuber, 1997) as the encoder and decoder. In order to solve the long-distance dependency problem in the encoding process, Wang et al. used the multi-head attention network Transformer (Vaswani et al., 2017) as the encoder and decoder, and achieved further performance improvement (Wang et al., 2019).

An ideal paraphrase not only needs to have the same semantics but also should have a significant change in expression from the input sentence (i.e., expression difference) (Bhagat and Hovy, 2013). Aiming at the problem of expression differences in generated sentences, researchers have made a lot of attempts in different dimensions (Lin and Wan, 2021; Li et al., 2019; Hosking and Lapata, 2021; Meng et al., 2021). For example, Li et al. proposed multiple generators with different granularity levels to learn the mapping relationship between input and output respectively, and then combine them to complete the paraphrase generation task (Li et al., 2019). Lin and Wan utilized back-translation and multiple rounds of iterative generation methods to produce paraphrased sentences with significant variance (Lin and Wan, 2021). Hosking and Lapata (Hosking and Lapata, 2021) continued to use the idea of variational autoencoder to encode the structure and semantics of the sentence separately, and generate paraphrases by perturbing the structure encoding. Different from these methods, this work learns to generate syntax-based prompts, which could induce the pre-trained model to generate diverse paraphrases.

Apart from the traditional methods working with language models (LMs) that have parameters less than 1B, modern LLMs like ChatGPT can also generate paraphrases with high quality. However, it costs much more than traditional methods since they require a huge training corpus and learnable parameters.

## 3 Method

### 3.1 Model Architecture

In this work, we propose VQPrompt, a novel model that generates paraphrases via prompt learning. It is composed of a prompt encoder and a pre-trained generative language model, which will be elaborated as follows.

### 3.1.1 Prompt Encoder

The prompt encoder aims to generate prompts for the pre-trained language model to produce reasonable paraphrases. The proposal of the prompt encoder stems from the assumption that the pre-trained language model (PLM) is powerful to generate sentences with arbitrary contents if given suitable inputs. Therefore, for a particular input sentence, the corresponding prompt is all we need for paraphrase generation in this work.

Since the prompts are dependent on the input sentence, this work introduces sample-aware prompt encoder. For a given sequence of tokens $\boldsymbol{x} = \{x_1, x_2, ..., x_n\}$, we first get the embeddings $\boldsymbol{e} = \{\mathbf{e_1}, \mathbf{e_2}, ..., \mathbf{e_n}\}$. Then we employ a sentence encoder to take the sentence embeddings as inputs and output the $M$ continuous prompts, given by

$$\boldsymbol{r} = \text{SentenceEncoder}(\boldsymbol{e}_1, \ldots, \boldsymbol{e}_n), \quad (1)$$

where $\boldsymbol{r}$ stands for the continuous prompt (with length of $M$) for the sentence $\boldsymbol{x}$. We adopt the encoder of the T5 model (Raffel et al., 2020) as the sentence encoder.

In general, the prompt in our work for paraphrase generation illustrates the abstract rule of paraphrase transformation. Indeed, humans have summarized several abstract rules of the transformation between paraphrases. For instance, the abstract rule "what is the reason of \$x? → why does \$x happen?" could characterize a number of transformations of paraphrases. Therefore, we expect that the prompt could indicate the abstract transforming rules for paraphrase generation.

Therefore, we make the second assumption that the transforming rules of paraphrase generation are finite. Based on the assumption, we propose a prompt encoder that produces discrete rule representations by vector quantization (VQ) (Zhang et al., 2022; Bhardwaj et al., 2022).

Formally, the prompt encoder maintains a codebook that comprises $K$ discrete prompt encodings. The above continuous prompt $\boldsymbol{r}_m$ are then quantized into discrete ones, selected from the codebook $\mathcal{C} \in \mathbb{R}^{K_c \times D}$, where $K_c$ is the number of the discrete code, and $D$ is the dimensionality of each discrete code $\mathcal{C}_k$. We measure the L2 distance between the continuous representation $\boldsymbol{r}$ and code vectors in the codebook. For each vector set $\boldsymbol{r} \in \mathbb{R}^{M \times D}$, where $M$ is the number of the continuous vectors and $D$ is the dimensionality of each continuous vector $\boldsymbol{r}_m$, the code vector that yields the minimum distance is taken to obtain the discrete rule

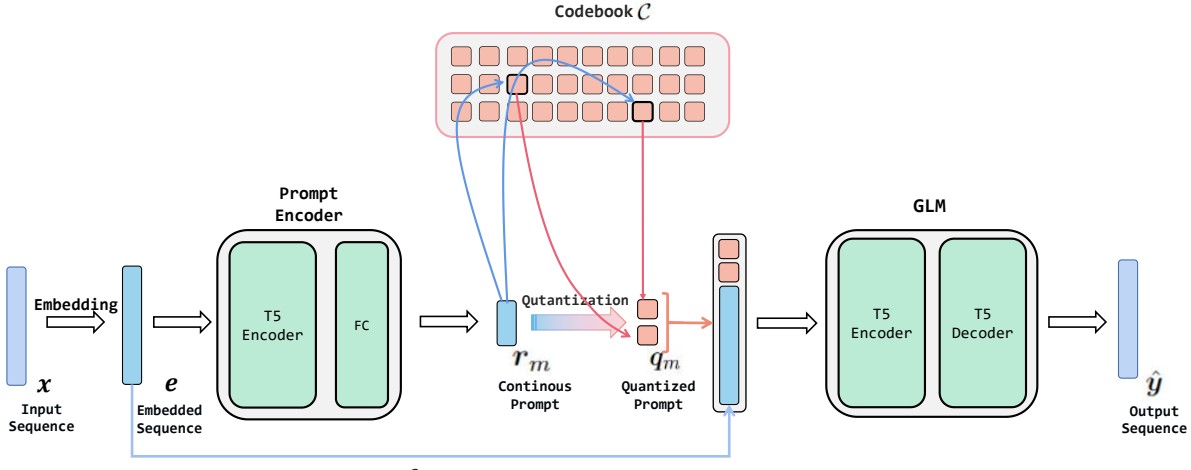

Figure 2: Model Architecture.

representation $q_m$. The detailed computations are

$$q_m = \mathcal{C}_k, \tag{2}$$

$$\text{where } k = \underset{j \in \{1,...,K_e\}}{\arg\min} \|r_m - \mathcal{C}_j\|_2,$$

where $q_m$ is the closest quantized vector for the continuous vector $r_m$. Finally, $M$ prompt vectors constitute a paraphrase prompt, given by

$$Q = \text{PromptEncoder}(e) = \{q_m | m = 1, \ldots, M\}, \tag{3}$$

where $Q$ is the final prompt generated by the prompt encoder for a particular sentence $x$.

To make the above vector quantization work, we need to train both the neural networks and the codebook in the prompt encoder toward the minimum of the distance between the continuous and discrete representations. The objective function of vector quantization for a particular data point is

$$\mathcal{J}^{\text{vq}}(x) = \|\text{sg}(r_m) - q_m\|_2^2 + \|r_m - \text{sg}(q_m)\|_2^2, \tag{4}$$

where $\text{sg}(\cdot)$ is stop-gradient operation during computation. In this way, we can derive the discrete rule representations, which are expected to be interpretable and instance-invariant.

Overall, the prompt encoder is a deep network attached with a discrete codebook $\mathcal{C}$, which is trained following the basic idea of VQ-VAE. The prompt encoder takes an embedded sequence e as input and generates several prompts $q_m$ as output, which contains syntactic structure information that guides the generative LM to produce paraphrases.

Note that the parameters of the generative language model in our work are fixed when we train the prompt encoder and the codebook. Therefore, the generative language model (LM) is neither

---

**Algorithm 1** K-means Update Algorithm

---
**Require:** Paraphrase dataset $\mathcal{D} = \{(x_n, y_n) | n = 1, \cdots, N\}$
1: Computing word embeddings $e_n = \textbf{EmbeddingLayer}(x_n)$
2: Collecting embeddings $E = \{e_n | n = 1, \cdots, N\}$
3: Initializing code list $C$
4: **for** e in E **do**
5:     $Q$ = Prompt Encoder($e$)
6:     **for** $m$ $in$ size($Q$) **do**
7:         $C$.append($q_m$)
8:     **end for**
9: **end for**
10: Obtaining codebook by computing the K-means centers of the code list $\mathcal{C}$ = **K-means**($C$)

**Ensure:** Codebook $\mathcal{C}$

---

learned to generate paraphrases nor able to capture the syntactic structure information of the target sentence. All of this information should be encoded by the prompt codes. That is to say, our work builds an information bottleneck where vector-quantized prompts are the only pathway to convey the syntactic structure information to the generative LM. In this specific and effective design, the acquisition of the syntactic information in the codebook can be well guaranteed.

### 3.1.2 Generative LM

A generative language model (LM) prescribes the generation of a sentence as a sequence of word predictions based on the context. The generative

LMs have made waves in the NLP community by demonstrating astounding few-shot capabilities on myriad language understanding tasks (Brown et al., 2020). Also, the generative language models possess with powerful decoding capacity; they could produce arbitrary contents if given suitable prompts. Therefore, the paraphrase generated by our model is given by

$$P(\cdot|\boldsymbol{x}) = \text{GLM}(\{Q \oplus \boldsymbol{e}\}) \qquad (5)$$

$$\hat{\boldsymbol{y}} \sim P(\cdot|\boldsymbol{x}) \qquad (6)$$

where GLM stands for the generative language model and the variable $Q$ means the sequence of prompts $\boldsymbol{q}_m$, i.e., $Q = \{\boldsymbol{q}_m | m = 1, \cdots, M\}$. $\oplus$ is the vector concatenation operation and $\hat{\boldsymbol{y}}$ is the generated sentence of VQPrompt.

This work aims to adapt the generative LM to produce paraphrases given the input sentence, which belongs to the task of conditional sentence generation. Therefore, we adopt an instruction-based language model named Flan-T5 (Chung et al., 2022) to serve as our base model. The fine-tuned language model (i.e., Flan-T5) takes a sequence of words as inputs and outputs several sentences (i.e., $\hat{\boldsymbol{y}}$) as needed.

### 3.2 Training Strategy

Similar to most paraphrase generators, VQPrompt is trained to fit the mapping from the input sentences to their paraphrases. Also, the paraphrase datasets are constructed as paraphrase pairs. Formally, let the dataset be $\mathcal{D}$ with size $N$. VQPrompt aims to maximize the log-likelihood (denoted by $\mathcal{J}^{LL}$) of the target paraphrases over all the training samples of $\mathcal{D}$, that is,

$$\mathcal{J}^{ML} = \sum_n^N \log P_\theta(\boldsymbol{y}_n|\boldsymbol{x}_n)$$

$$= \sum_n^N \sum_t^{T_n} \log P_\theta(y_{n,t}|\boldsymbol{y}_{n,<t}, \boldsymbol{x}_n), \qquad (7)$$

where $y_{n,t}$ stands for the $t$-th word of the target paraphrase in the $n$-th sample. $T_n$ denotes the word length of the target paraphrase $\boldsymbol{y}_n$. $\theta$ is the model's parameters.

Together with the objective of VQ, the final objective function $\mathcal{J}$ of VQPrompt is

$$\mathcal{J} = \mathcal{J}^{ML} + \sum_n^N \mathcal{J}^{\text{vq}}(\boldsymbol{x}_n) \qquad (8)$$

However, the parameters of the prompt encoder are difficult to optimize since the vector quantiza-

| Dataset | #Train set | #Val set | #Test set |
|---------|-----------|----------|-----------|
| Quora   | 55,611    | 5,255    | 5,255     |
| Paralex | 222,223   | 27,778   | 27,778    |
| MSCOCO  | 113,287   | 5,000    | 5,000     |

Table 1: Statistics of the benchmark datasets used in this work.

tion intercepts the gradients of backpropagation. Our preliminary experiments reveal that most of the codes in the codebook are rarely selected by the prompt encoder after the optimization based on the objective function $\mathcal{J}$, which is called index collapse (Wu and Flierl, 2020). The index collapse usually happens on text generation since its gradients are not smooth enough.

Therefore, we propose a new training strategy (called K-means training) to eliminate the index collapse in the prompt encoder. K-means training contains the following two stages:

**Codebook warm-up.** We first ignore the codebook of the prompt encoder and directly use the continuous prompt to perform the paraphrase generation. Thus, the training objective is only to minimize the maximum likelihood objective $\mathcal{J}^{ML}$.

**K-means Update.** Before the training in this stage, We sample some sentences from the dataset and collect the corresponding prompt codes generated by the randomly initialized VQPrompt model. Then we perform the K-Means algorithm on those codes and collect a set of codes as the primitive version of the codebook. Next, during the training, we prevent index collapse by updating the dead codes in the codebook with the clustered center. When the amount of active codes is lower than a threshold $\mathcal{T}$, we will perform the replacement. If the codes are not used for a relatively long time when training, we say that the code is dead.

**Discussion of K-means Training.** In essence, the K-means strategy is an update trick in the optimization of the prompt encoder. However, the index collapse in VQ has been a long-standing problem in deep generative modeling (Łańcucki et al., 2020; Wu and Flierl, 2020). The proposed K-means strategy works well empirically and has the potential to benefit other vector quantization models. But figuring out the underlying theory is nontrivial, which we leave as future work.

## 4 Experiments

In this section, we first test the proposed model on the benchmarking datasets on both automatic

| Model | Quora | | | Paralex | | | MSCOCO | | |
|---|---|---|---|---|---|---|---|---|---|
| | BLEU | self-BLEU | iBLEU | BLEU | self-BLEU | iBLEU | BLEU | self-BLEU | iBLEU |
| Copy | 34.52 | 100.00 | 7.61 | 37.10 | 100.00 | 9.68 | 19.85 | 100.00 | -4.12 |
| tf-idf | 24.05 | 62.49 | 6.75 | 25.08 | 25.25 | 15.01 | 18.26 | 38.37 | 6.93 |
| AE | 28.99 | 60.11 | 11.17 | 40.10 | 75.71 | 16.94 | 27.90 | 38.71 | 14.58 |
| VAE | 27.23 | 51.09 | 11.57 | 38.91 | 53.28 | 20.47 | 27.44 | 24.40 | 16.99 |
| VQ-VAE | 16.31 | 21.13 | 8.83 | 40.26 | 65.71 | 19.07 | 25.62 | 22.41 | 16.01 |
| SOW/REAP | 21.27 | 38.1 | 9.41 | 33.09 | 37.07 | 19.06 | 12.51 | 6.47 | 8.71 |
| BTmPG | 19.83 | 35.11 | 8.84 | 28.40 | 35.99 | 15.52 | 19.76 | 13.04 | 13.20 |
| LBoW | 23.51 | 42.08 | 10.39 | 34.96 | 35.86 | 20.80 | 21.65 | 16.46 | 14.02 |
| Separator | 23.68 | 24.20 | 14.10 | 36.36 | 35.37 | 22.01 | 20.59 | 12.76 | 13.92 |
| HRQ-VAE | 33.11 | 40.35 | 18.42 | 39.49 | 33.30 | 24.93 | 27.90 | 16.58 | 19.04 |
| VQPrompt-PG | 35.01 | 39.98 | **20.01** | 42.58 | 41.96 | **25.67** | 29.92 | 23.59 | **19.21** |

Table 2: Performance of individual paraphrase generation methods on the Quora and Paralex, and MSCOCO datasets.

evaluation and human evaluation. Then we provide several detailed analyses to elucidate its effectiveness in generating paraphrases.

## 4.1 Datasets

In this work, we use three widely used benchmarking datasets, namely, Quora (Chen et al., 2017), MSCOCO (Lin et al., 2014), and Paralex (also named Wiki Answers) (Fader et al., 2013) in our experiments.

**Quora**. The Quora dataset is collected from the question-answering forum Quora (Chen et al., 2017). It contains over 400k pairs of questions, some are paraphrases and others are non-paraphrases. There are about 150k paraphrase pairs in total.

**Paralex**. Paralex is a dataset of question paraphrases datasets scraped from WikiAnswers (Fader et al., 2013). It has a large number of question pairs but presents lower quality in syntactic structures and semantic similarity compared to Quora.

**MSCOCO**. MSCOCO is a benchmark dataset for image captioning (Lin et al., 2014). It contains over 100k clusters of five captions sentences. Considering captions for images can involve different details or objects, the quality of these paraphrases is lower than those in Quora.

For the fairness of comparison, We use the cluster version of these three datasets released by the previous best method (i.e., HRQ-VAE (Hosking et al., 2022)). The statistics of the training, validation and test splits are shown in Table 1.

## 4.2 Competing Methods

We will compare VQ-prompt with multiple advanced paraphrase generation models. We describe several most competing models as follows.

**SOW/REAP**. It uses a two-stage model to derive a set of syntactic rearrangements, which are then used to guide an encoder-decoder model (Goyal and Durrett, 2020).

**BTmPG**. It leverages a multi-round paraphrase generator to improve diversity and back-translation to preserve semantic information (Lin and Wan, 2021).

**LBoW**. It grounds the semantics of a discrete latent variable by the latent bag-of-words technique (LBoW) (Fu et al., 2019).

**Separator**. (Hosking and Lapata, 2021) take both the semantic sentence and syntax-informed sentence as inputs in the training process. It combines training objective with a principled information bottleneck, to induce a latent space that disentangles meaning and form.

**HRQ-VAE**. Hierarchical refinement quantized variational autoencoders (HRQ-VAE) is a method for learning the decomposition of dense encodings as a sequence of discrete latent variables that make iterative refinements of increasing granularity (Hosking et al., 2022). HRQ-VAE serves as the previous state-of-the-art paraphrasing method. We take it as our arch-rival.

## 4.3 Evaluation Metrics

Many previous works adopt BLEU as a measure for evaluating several text generation tasks. But for paraphrase evaluation, the dissimilarity from

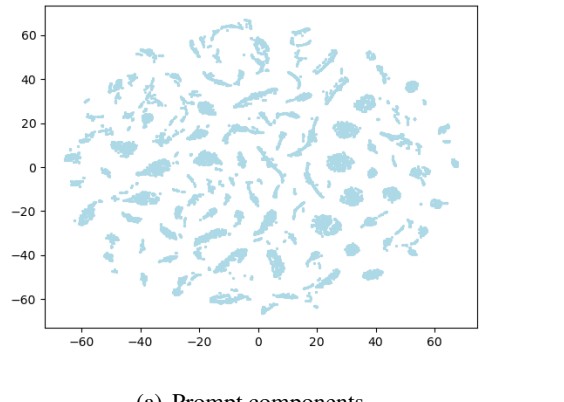

(a) Prompt components

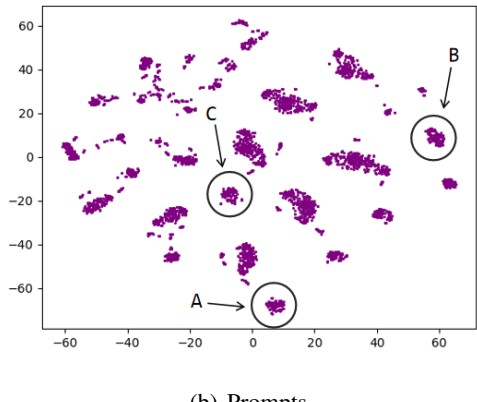

(b) Prompts

Figure 3: Visualization of the learned prompts and their components.

the input is also of vital importance. So, in order to take both paraphrase quality and similarity to the input into consideration, we also use iBLEU for our automatic evaluation. The calculation of iBLEU is given by

$$\mathbf{iBLEU} = \alpha \cdot \mathrm{BLEU}(\hat{\boldsymbol{y}}, Y) -$$
$$(1 - \alpha) \cdot \mathrm{BLEU}(x, Y) \quad (9)$$

where $Y$ stands for the set of reference paraphrases. Thus, the expression $\mathrm{BLEU}(x, Y)$ indicates the BLEU score between the input sentence and the reference paraphrases, which is also called the Self-BELU score. The coefficient $\alpha$ balances the importance between expression diversity and semantic similarity. Following the setting of (Hosking et al., 2022), we set $\alpha = 0.8$.

Overall, the BLEU, Self-BLEU, and iBELU scores constitute a relatively comprehensive evaluation of the generated paraphrases. In addition to the automatic evaluation metric, we also conducted the human evaluation.

### 4.4 Implementation Details

The hidden layer sizes of the equation encoder and the expression generator are 768. The size of the codebook is set to 512. The length of the prompt (i.e., $M$) is 4. The threshold $\mathcal{T}$ in the K-means training strategy is 256. The maximum input length of the feature vector is 256 and the maximum output length is 60. We evaluate the model for each half epoch and select the model that reaches the best performance on the validation set. Finally, we report the generation performance on the test set.

| Model | Semantic relevance | | Fluency | |
|---|---|---|---|---|
| | Mean Score | Variance | Mean Score | Variance |
| Separator | 1.94 | 0.30 | 0.60 | 0.35 |
| HRQ-VAE | 2.22 | 0.67 | 2.84 | 0.17 |
| VQ-Prompt | **2.32** | 0.61 | **2.90** | 0.10 |

Table 3: Human evaluation.

### 4.5 Results

Table 2 presents the performance of all competing methods on the Quora, Paralex, and MSCOCO datasets. Copy and tf-idf are typically repeating the input sentences and thus obtain the lowest iBLEU scores. The neural networks, including LBoW, VAE, and SEPARATOR, achieve higher iBLEU scores. But these improvements are obtained with the loss of the semantic meaning because the similarity with the references is decreased along with the improvements of iBLEU. HRQ-VAE is the previously state-of-the-art paraphrase generator, which obtains better performances than SEPARATOR and LBoW. However, HRQ-VAE prescribes that the dataset contains high-quality sentence pairs with similar syntax structures, which is not feasible in sentences with complex grammar dependence.

As for VQPrompt, we observe that it consistently outperforms HRQ-VAE and the other baselines on the three benchmark datasets. Considering that HRQ-VAE utilizes additional syntax supervision, the improvements on both BLEU and iBLEU demonstrate the effectiveness of the proposed method.

**Human Evaluation.** We also conducted a human evaluation of the results. Due to the limit of budget and resources, we sampled 300 sentences from

| Cluster | Input/Output | Sentence |
|---|---|---|
| A | Input | How can I learn to speak Spanish fluently? |
| | Generation | What is the best way to learn Spanish? |
| | Input | How can I lose 25 pounds in one month in a safe way? |
| | Generation | What are some ways to lose 25 pounds in 1 month? |
| | Input | How can you substitute tarragon in recipes? |
| | Generation | What are some ways to substitute tarragon in recipes? |
| B | Input | What should you do to prepare for the RCMP? |
| | Generation | How can I prepare for the RCMP? |
| | Input | What should I follow to keep myself fit without going to gym? |
| | Generation | How can I stay fit without going to a gym? |
| | Input | What should I do to make life worth living? |
| | Generation | How can I make my life worth living? |
| C | Input | What makes a dog's vomit foamy? Is it dangerous? |
| | Generation | Why is my puppy throwing up yellow liquid? |
| | Input | What causes tides to rise and fall? |
| | Generation | Why do the tides in the sea rise? |
| | Input | What is the story behind Obama abstaining from the vote on Israeli? |
| | Generation | Why did President Obama abstain in the UN vote against Israeli? |

Table 4: Paraphrases generated from the prompt clusters shown in Fig 3(b).

| Model | BLEU | Self-BLEU | iBLEU |
|---|---|---|---|
| Generative LM | 34.27 | 42.75 | 18.87 |
| Generative LM (VQ) | 32.51 | 46.04 | 16.80 |
| VQPrompt | 35.01 | 39.98 | **20.01** |

Table 5: The generation performances of individual VQPrompt variants.

the Quora test set and compared VQPrompt with Separator and HRQ-VAE. We asked three human annotators to evaluate the generated paraphrases in terms of semantic relevance and sentence fluency in a blind fashion; each aspect was scored from 1 to 5. We report in Table 3 the average human scores and their variances. Table 4 shows that VQPrompt achieves the highest human satisfaction scores. The results are also consistent with the automatic metrics in Table 2.

**Ablation Study.** In order to investigate the reasons for the performance gain obtained by VQPrompt, we further build two variants of VQPrompt and evaluate their generation results. The two variants are the generative language model (denoted by generative LM) and the generative language model with traditional vector-quantized prompt (generative LM (VQ)). The difference between generative LM (VQ) and the proposed VQPrompt model lies in the optimization of the prompt encoder (i.e., the K-means training strat-

egy). These two variants together with VQPrompt share the same hyperparameters and data.

As the Quora dataset is the most widely-used high-quality dataset, the ablation study is only conducted on Quora. As shown in Table 5, the generative LM model reaches a modest performance, owing to the decent initialization of the pre-trained language model. Next, simply adding a discrete prompt to the model would lead to a side effect of the paraphrase generation, which is caused by the index collapse of the VQ technique. With our training scheme, the discrete presentation of prompts could further boost the performance of the generative LM. We also observe that more than half of the codes in the codebook are active after incorporating the training scheme, which reflects the VQ computations works well and can finally benefit the paraphrase generation.

**Prompt Visualization.** For an intuitive visualization of generated prompts, we perform t-SNE on prompt component codes $q_m$ and the prompts $Q$. In this paper, $M$ component codes constitute a paraphrasing prompt (in experiments, $M = 4$). Although we use the same number of vectors to conduct t-SNE, the dimension reduction results are varied. Generally, the points of the paraphrase prompt tend to clump together in larger clusters, indicating that VQPrompt has learned several ab-

stract paraphrasing rules, which could induce the pre-trained model to produce paraphrases.

To demonstrate this point, we select three clusters and use them to perform paraphrasing. As shown in Table 5, we observe that these clusters contain a bunch of sentences that share similar syntactic structures, which validates that the learned prompts characterize the abstract transforming rule of paraphrase generation.

## 5 Conclusion

Paraphrasing aims to restate one sentence as another with the same meaning, but different wordings. In this paper, we establish a prompt learning framework, coined VQPrompt, for paraphrase generation. VQPrompt leverages vector quantization to learn finite prompt components and thus possess modest interpretability. We introduce a K-means training strategy to avoid index collapse in VQ. Experiments show VQPrompt achieves impressive generation performance on multiple datasets.

## 6 Limitations

For ethnic concerns, the three datasets we use are publicly available and do not contain biased or discriminatory information. For resource concerns, our model is dependent on a pre-trained model, which means a higher computation budget.

## Acknowledgements

This work was supported by the National Natural Science Foundation of China under Grant 62206192, the Natural Science Foundation of Sichuan Province under Grant 2023NS-36 FSC1408 and the MIIT Industrial Internet Innovation and Development Project.

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
