# OpenReview forum: "Vector-Quantized Prompt Learning for Paraphrase Generation"
_EMNLP/2023/Conference — EMNLP 2023 Findings_

### Official Review · Reviewer_CUyp · 2023-08-05

**Soundness:** 3

**Excitement:**

2: Mediocre: This paper makes marginal contributions (vs non-contemporaneous work), so I would rather not see it in the conference.

**Missing References:**

[1] Rishabh Bhardwaj, Amrita Saha, Steven C.H. Hoi, and Soujanya Poria. 2022. Vector-Quantized Input-Contextualized Soft Prompts for Natural Language Understanding. EMNLP 2022.

**Paper Topic And Main Contributions:**

This paper presents a method to improve paraphrase generation using prompts. The main assumption is that the paraphrase generation has limited transformation formats, which can be captured by a small number of discrete vectors through quantization. During generation, an encoder model is first applied to obtain M continuous prompts based on current input, which are later used as queries to find
a set of relevant quantized vectors stored in a codebook. An encoder-decoer model takes the concatenation of the retrieved quantized vectors and the input sentence as the input to generate the desirable paraphrased sentence. In order to make sure the proposed discrete vectors can be learned jointly with other parts of the model, the author also employs an online k-means algorithm to update
the quantized vectors. The results on three benchmarks show that the proposed method outperforms a variety of baselines according to both automatic evaluation metrics and human evaluation. The ablation analysis also verifies the effectiveness of each proposed component. The case study also shows that some of the learned quantized prompts can encode information related to some specific types of transforming rules.

**Questions For The Authors:**

1. In Eq.1, how can you get a continuous prompt with length M by taking a sentence with length n? Do you mean adding M tokens in front of the input sequence for prompt encoding?
2. How the number of discrete vectors in the codebook affects the performance is not clear.
3. It's not fair to compare the proposed model with other methods, as it uses Flan-T5 for model initialisation. Have you tried other pre-trained models, such as those used in HRQ-VAE.

**Reasons To Accept:**

1. The proposed quantized-prompt method improves the quality and diversity of paraphrase generation empirically.
2. The motivation behind using quantized-prompts to encode paraphrase rules is reasonable.

**Reasons To Reject:**

1. This paper shares similar ideas with Vector-Quantized Input-Contextualized Soft Prompts for Natural Language Understanding (Bhardwaj et al., 2022), in terms of both model architecture and employing k-means algorithm for quantized-prompt learning. How these two works differ is unclear, which limits the novelty of this paper.
2. The author mentioned that the learned prompts could encode syntax-related information, but this claim is not supported well in experiments.
3. The author argues that a limited number of quantized prompts are able to encode the information related to transforming rules for paraphrase generation, which should not be dedicated to a specific dataset. Thus, we have a concern that how well the quantized prompts learned in one dataset or one domain performs in the unseen domains. However, this essential property was not examined.

**Reproducibility:**

3: Could reproduce the results with some difficulty. The settings of parameters are underspecified or subjectively determined; the training/evaluation data are not widely available.

**Reviewer Confidence:**

3: Pretty sure, but there's a chance I missed something. Although I have a good feel for this area in general, I did not carefully check the paper's details, e.g., the math, experimental design, or novelty.

**Typos Grammar Style And Presentation Improvements:**

line 200: "a For" -> Delete "a"

lines 172-177: "Hosking and Lapata continues to ..... (Hosking and Lapata, 2021)" -> "Hosking and Lapata (2021) continues to ....."

---

> ### Author Rebuttal · Authors · 2023-08-29
>
> First of all, we sincerely appreciate the reviewer CUyp for your helpful feedback. And we carefully made the following point-to-point responses:
>
> **Q1. How to get a continuous prompt?**
>
> We first pad every input sentence to a fixed length (which is set to be 64 in our experiment). Then We feed the input into the prompt encoder and compress it to a prompt with length M by an MLP.
>
> **Q2.Influence of the number of discrete vectors in the codebook.**
>
> The codebook is set to 256 and the number of prompts is set to 4 on all the datasets. Although the size of the codebook is not large, the combination space of the four codes is huge. Also, we find that increasing the codebook size from 256 does yield better performances. Therefore, the size of the codebook is dependent on the complexity of the paraphrasing problem, and a size of 256 is enough for most of the paraphrasing tasks.
>
> **Q3. Not fair to compare the proposed model with other methods.**
>
> The previous SOTA method, HRQ-VAE uses the embedding layer of Bert, which is also a powerful pre-trained model. In addition, to fairly compare our method with HRQ-VAE, we also only use the embedding layer of our pre-trained model, and other parameters are randomly initialized. As shown in the following table, our method still outperforms HRQ-VAE by a noticeable margin.
> |  Method	   |  Bleu  | Self-bleu | iBleu |
> |  ----  | ----  | ----  |----  |
> | HRQ-VAE  | 33.11 | 40.35 | 18.42|
> | VQPrompt (random init.) | 32.40 | 33.83 | 19.15|
>
> By the way, HRQ-VAE has relatively high demands on the dataset, as it uses additional information about the syntax structure as inputs. Even in this condition, our method is still shown to be more effective.
>
>
>
> ## Response to the weakness
> **Q4. Similarity to Bhardwaj et al., 2022.**
>
> Our method is different from Bhardwaj et al., 2022 for the following reasons:
>
> (1) We use instance-dependent quantized prompts to produce a finite set of transforming prompts, which also aims to reduce the negative effect of noise during training as the training data contains some records of low quality.
>
> (2) The usage of K-means during training is not the same. We record the output of the sentence of the encoder in each epoch and cluster all the codes recorded. The center of each cluster will become the code in the new codebook.
>
> We will cite this paper and discuss the difference in the revised paper considering the rigor of the research.
>
> **Q5: Syntax-related information in the learned prompts.**
> The evidence of our prompt learning syntax-related information can be found in Table 5. We notice that different prompt clusters can perform different transformations on input sentences, but prompts in the same cluster produce sentences with similar syntactic structures.
>
> **Q6: The generalization ability of quantized prompts.**
>
> As shown in Table 5 in the manuscript, the learned quantized prompts could characterize the syntax-related information. These prompts are generalizable to the other domains if the target domains share similarities with the training data. To demonstrate it, we build a new paraphrasing dataset for the evaluation of the generalization ability. We leverage ChatGPT to produce diverse paraphrases and ask native speakers to refine these samples. Finally,  500 high-quality samples are obtained, each containing four paraphrases. We will make them open-source in the future. We observe that our method VQPrompt surpasses the previous SOTA method HRQVAE by a large margin, indicating the better generalization ability of our method.
>
> |  Method	   |  BLEU| Self-BLEU| iBLEU|
> |  ----  | ----  | ----  |----  |
> | HRQ-VAE  | 12.84 | 27.52 | 4.76|
> | VQPrompt | 19.51 | 20.63 | 11.48|
>
> **Q6: Typos**
> We will further check the presentation and typos in the revised manuscript. Thanks for your careful inspection.

---

### Official Review · Reviewer_j1cc · 2023-08-09

**Typos Grammar Style And Presentation Improvements:** 1. Please provide more detailed Figur…
**Soundness:** 4

**Excitement:**

4: Strong: This paper deepens the understanding of some phenomenon or lowers the barriers to an existing research direction.

**Paper Topic And Main Contributions:**

In this paper the authors propose a new model VQPrompt for paraphrase generation based on pre-trained model T5 along with vector quantized training strategy. The paper improves on traditional VQ training by using a novel K-means update which prevents index collapse. On three benchmark datasets and via automatic and human evaluation the proposed model is shown to perform better than state-of-the-art methods in paraphrase generation.

**Questions For The Authors:**

1.	What is the size of T5 model used for prompt encoding and GLM ?
2.	It is unclear how various parameters such as size of codebook K, number of prompts M chosen? Do the authors perform hyper-parameter tuning? What range of hyper-parameters were considered?
3.	Why not include the fine-tuned T5 based paraphrase generator (which I believe is referred as Generative LM in Table 4) in the main Table 2 itself since its is a very relevant baseline?


**Reasons To Accept:**

1.	The paper develops a novel paraphrase generation model based on pre-trained T5 and vector quantization of the intermediate prompt/ states and also provides a reasonable justification of why it works well
2.	The paper proposes a new k-means based approach for training VQPrompt model which avoids index collapse
3.	The VQPrompt model is thoroughly evaluated on benchmark paraphrase datasets and achieves SOTA results. Human evaluation is also performed to validate the quality of the generated paraphrases


**Reasons To Reject:**

1.	The writing can be improved. Some notations in Section 3 are not very clear. From equation 1, $r$ is the SentenceEncoding of the given input $x$. What is the dimension of $r$, is it a single vector or a collection of $M$ vectors of certain dimension? In Line 233, the authors say $r$ is of dimension $M \times D,$ but refers is as a vector. This is confusing, since if $r$ represents a collection of vectors, it should be referenced as a matrix or a set, not vector. I suggest using upper case $\mathcal{R}$ to represent matrix or set of vectors. Also the notation $\mathbb{R}^{M \times D}$ mathematically refers the field of $M \times D$ matrices over real numbers. Also how do you transform Sentence embedding of $n$ sequences into a set of continuous $M$ vectors? Figure 2 shows a FC layer, but it is not expressed mathematically.
2.	The parameter sizes of various baseline models are not clear. Does the VQPrompt model use separate T5 encoders for prompt encoding and the GLM? This considerably increases the model size.
3.	Due to the intermediate K-means step, and stop gradients, the training stability is questionable. Can the authors comment on this? The paper doesn't mention how many epochs of training are needed and the learning rates used


**Reproducibility:**

3: Could reproduce the results with some difficulty. The settings of parameters are underspecified or subjectively determined; the training/evaluation data are not widely available.

**Reviewer Confidence:**

3: Pretty sure, but there's a chance I missed something. Although I have a good feel for this area in general, I did not carefully check the paper's details, e.g., the math, experimental design, or novelty.

---

> ### Author Rebuttal · Authors · 2023-08-29
>
> First of all, thanks for your valuable and constructive suggestions. And we carefully made the following point-to-point responses:
>
>
> **Q1: Size of T5 model**
>
> The T5 model used in our work is T5-Flan-Base, which has 0.25B parameters. GLM is the intact T5-Flan-Base model, and the prompt encoder in our model only adopts the encoder part of T5-Flan-Base. And they do not share parameters.
>
>
> **Q2: Hyperparameters**
>
>  The codebook is set to 256 and the number of prompts is set to 4. We do not perform hyper-parameter tuning. But we add a set of experiments that shows the performance of our model when the number of prompts M varies. Below is the result.
> | Prompt length		| BLEU| Self-BLEU| iBLEU|
> |  ----  | ----  |----  | ----  |
> |1  |      33.30  |  34.26  |   18.79|
> |2	| 33.32 |	33.78 |	19.10|
> |4	|			35.01	|39.98|	20.01|
>
>
> **Q3: including the fine-tuned T5 as baseline**
>
> Actually, the performance of the fine-tuned T5 paraphrase generator is already reported in Table 4, which is considered to be a part of our ablation study. We will further clarify this point in the revised paper.
>
> ## Rebuttals for reject reasons:
>
> **Q4. The writing can be improved.**
>
> Sorry for causing confusion. R is actually a vector set that contains an M vector of dimension D. We will rectify this error in our paper.
>
> As for how we transform the input sentence of variable length to fixed-length vectors, we first pad every input sentence to a fixed length (which is set to 64 in our experiment). Then We feed the input into the prompt encoder and compress it to a prompt with length M by an MLP. We will clarify this point in the revised manuscript.
>
> **Q5. Parameter sizes of models.**
>
> The T5 model used in our work is T5-Flan-Base, which has 0.25B parameters. GLM is the intact T5-Flan-Base model, and the prompt encoder in our model only adopts the encoder part of T5-Flan-Base. The model sizes of VQPrompt and HRQ-VAE are similar, which are around 0.3B. In the revision, we will indicate the model sizes of all the methods.
>
> **Q6. Hyperparameters.**
>
> For our experiment reported in Table 2, we train for 16 epochs and select the best one. The learning rate is set to be 0.0005 and the batch size is 48 * 10 (accumulation steps) = 480. Using the proposed techniques (such as K-means clustering), the training process of VQPrompt is stable. We will release all the code once our submission is accepted.
>
> **Q7. Presentation Improvements.**
>
> We will provide more detailed captions in Figures 2 and 3 to explain the idea of our method. Thanks for your suggestion.

---

### Official Review · Reviewer_F6VU · 2023-08-09

**Soundness:** 3

**Excitement:**

4: Strong: This paper deepens the understanding of some phenomenon or lowers the barriers to an existing research direction.

**Paper Topic And Main Contributions:**

The paper addresses the challenges in generating diverse and high-quality paraphrases by leveraging the power of pre-trained language models and instance-dependent prompts. It introduces a vector-quantized strategy to encode prompts and generate paraphrases using vector-quantization using a codebook of prompt templates which dynamically updates itself using k-means based on an explicit assumption that the abstract transforming patterns of paraphrase generation are finite and not numerous. They use the T5-encoder and the Flan-T5 models (text-to-text encoder-decoder). The research evaluates its proposed model, VQPrompt, on four paraphrasing datasets: Quora, Paralex (WikiAnswers), and MSCOCO, comparing it with several advanced paraphrase generation models. Their experimentation shows claimed sota on the paraphrasing tasks.

**Questions For The Authors:**

Q1a. Considering the current dominance of modern generative language models in paraphrasing tasks, how might the proposed VQPrompt framework compare in terms of efficiency and performance when applied to closed LLMs like GPT models and open LLMs? Could the paper explore potential trade-offs between VQPrompt's performance and inference costs, shedding light on the practicality of its utilization?

Q1b. While the paper acknowledges concerns regarding the scalability and generalization of the VQPrompt model due to its limited evaluation on benchmark datasets, how might the proposed framework handle more diverse and challenging real-world paraphrasing tasks, different domains, and languages? Could the authors provide insights into addressing domain adaptability and potential concept drift to enhance the model's robustness and applicability beyond the evaluated datasets?

Q1c. Lastly, what other applications of this vector-quantization style approach might be done? It seems interesting to port it to other use-cases where current generative LLMs might have huge inference costs.

**Reasons To Accept:**

## Strengths

1. **Interesting Approach:** The use of a vector-quantization method using a codebook and using k-means to control the codebook seems like an interesting approach that I have not personally come across before even for other tasks.

2. **Potential for Generalization and Scaling:** The project's approach of utilizing pre-trained language models and incorporating them into the paraphrase generation process using vector quantization has the potential to be applied to other text generation tasks beyond paraphrasing, contributing to the broader field of natural language generation. This seems like an interesting way to prompt on more types of language models for more tasks which might have rule-based idiosyncracies to them.

3. **Effective use of LMs T5 and Flan-T5**: The paper leverages pre-trained language models and demonstrates how they can be adapted for the task of paraphrase generation.

4. **Interpretability:** The paper emphasizes the interpretability of the learned prompts, which is an important aspect of the proposed framework. The discrete rule representations obtained through vector quantization offer a degree of transparency and insight into the paraphrase generation process.

5. **Reported SOTA:** The experimental results demonstrate that VQPrompt achieves new state-of-the-art performance on benchmark datasets, including Quora, Wikianswers, and MSCOCO.

6. **Formulas and Pseudo code:** I like that the paper has well presented formulas and pseudo-code.

**Reasons To Reject:**

## Weaknesses

1. **Complex Presentation & Writing Style:** Although the approach seems to be executed well, I feel the writing style could be more well structured and a better insight as to the motivation of the approach could have been given. I personally did not have much experience with vector-quantization methods and complex nuances such as the index collapse issue and more such things were initially difficult to follow. A well-rounded understanding for this should have been presented.

2. **Efficiency wrt current generative LLMs:** I also would have been interested in observing the performance of this approach when applied to modern generative LLMs, given their current superiority in paraphrasing tasks. It would have been valuable to have comparisons made between VQPrompt and both closed LLMs like the GPT models and the open LLMs. Even if the VQPrompt framework didn't directly outperform them, analyzing the trade-off between performance and inference costs on the Pareto front would have been insightful.

3. **Vector Quantization may Backfire?:** The paper evaluates the proposed approach on a limited set of benchmark datasets, which may not fully represent the diversity and complexity of real-world paraphrasing tasks. The lack of evaluation on more diverse and challenging datasets raises concerns about the scalability and generalization of the VQPrompt model to different domains and languages. End-to-end text-to-text paraphrasing might be more generalizable on a myriad of text types, and I have concerns related to domain adaptability and concept drift in the future.

- It appears that the solution space for this task may be large. Better Auto-Encoder based models which are more robust might outshine this approach, changing the generative decoding methods such as using beam-search trees or top sampling might generate the necessary valid perturbations needed (although I'm not sure how the outputs would compare). Beam search can help improve the diversity of generated paraphrases by considering multiple candidate sequences during decoding; this could enhance the model's ability to produce a wider range of paraphrase variations, modifying the decoding sampling technique, such as using nucleus sampling or temperature scaling, can provide more control over the level of diversity in the generated paraphrases.

4. **Not Enough Discussion of Negative Scenarios:** The paper mainly highlights the positive results and state-of-the-art performance achieved by VQPrompt. However, the absence of a discussion on scenarios where the proposed approach might fail or produce suboptimal results limits the paper's comprehensive analysis and practical guidance for potential users.

5. **Codebook Maintenance and Trade-offs:** I'm not sure how the codebook would be updated with time and concept drift scenarios. A more detailed analysis of the trade-offs and computational costs associated with codebook maintenance is needed. Would like to see a more critical assessment of statements made.

6. **Minimal Discussion of Limitations**: The limitations section is negligible.

**Reproducibility:**

3: Could reproduce the results with some difficulty. The settings of parameters are underspecified or subjectively determined; the training/evaluation data are not widely available.

**Reviewer Confidence:**

3: Pretty sure, but there's a chance I missed something. Although I have a good feel for this area in general, I did not carefully check the paper's details, e.g., the math, experimental design, or novelty.

---

> ### Author Rebuttal · Authors · 2023-08-29
>
> We thank the reviewer for the instructive feedback. We will first answer the questions, and then we will respond to the weakness.
>
> **Q1: Comparison between VQPrompt with LLMs.**
>
> Following the suggestion of the reviewer, we evaluate the efficiency and performance of the closed/open LLMs. ChatGPT is the most advanced closed LLMs and the Vicuna is an open-source LLMs finetuned on the LLaMA model. As shown in the following table, our method VQPrompt outperforms both ChatGPT and Vicuna on the Quora dataset. After analyzing the generation, we observe that the paraphrases generated by the ChatGPT and Vicuna are reasonable but still similar to the original sentence, thus yielding to lower iBLEU score. When we fine-tune the Vicuna with the training samples of Quora, Vicuna exhibits the best performance, indicating the generative power of LLMs. However, LLMs consume huge computational resources but bring marginal performance gains. Therefore, the proposed method, as the trade-off of the computational costs and the performance, stands as an effective paraphrase generator.
> |  Models	| Type	  | Parameters	  | iBLEU |
> |  ----  | ----  | ----  | ----  |
> | Vicuna	  | Open source | 70B	| 10.21 |
> | Vicuna (fine-tuned) | Open source | 70B	|21.11|
> |ChatGPT|Closed source |>175B|10.41|
> | VQPrompt| Open source |0.25B|20.01|
>
> **Q2: generalization ability**
>
> To conduct experiments on more diverse and challenging paraphrasing tasks is a good idea, but there does not exist such a dataset that satisfies these requirements. The current benchmark datasets, such as Quora and Wiki, are collected from different domains and do not share reasonable semantics, which forbids the model from learning a transferable ability to paraphrase.
>
> Therefore, in this work, we build a new paraphrasing dataset for the evaluation of the generalization ability. We leverage ChatGPT to produce diverse paraphrases and ask native speakers to refine these samples. Finally, 500 high-quality samples are collected, each containing four paraphrases. We will make them open-source in the future. We observe that our method VQPrompt surpasses the previous SOTA method HRQVAE by a large margin, indicating the better generalization ability of our method.
>
> |  Method	   |  BLEU| Self-BLEU| iBLEU|
> |  ----  | ----  | ----  |----  |
> | HRQ-VAE (random init.) | 12.84 | 27.52 | 4.76|
> | VQPrompt | 19.51 | 20.63 | 11.48|
>
>
> **Q3: Application of vector-quantization to LLMs.**
>
> R1c: In fact, our vector-quantize technique is used to build a finite set of transforming prompts, which has the potential to reduce the negative effect of noise during training and produce better performance and interpretability. It’s different from the quantized technique used in accelerating the inference of LLMs. We will clarify this point in the revised paper.
>
> ## Responses to [Weakness]
>
> **Q4: Complex presentation & writing style.**
>
> We are encouraged that the reviewer comments that our approach is well executed. We will explain the proposed techniques such as vector quantization and index collapse in more detail.
>
> **Q5. Vector quantization may backfire**
>
> Admittedly, the expressive capacity of the vector quantization may be limited as the sacrifice to obtain the interpretability. However, except for the vector quantization, most of the parts in VQPrompt are continuous, which are highly non-linear, and capable of learning to fit the diversity and complexity of real-world paraphrases. Therefore, the influence of the vector quantization on the fitting power of our model is small and can be ignored.
>
> In addition, as stated in the responses to Q2, the generalization ability of VQPrompt is stronger than the previous SOTA models.
>
> **Q6. Negative Scenarios and Limitations**
>
> We thank the reviewer’s suggestions to further discuss the negative scenarios and the limitations of our method. In the submission phase, we are not able to provide more analyses on the main text due to the page limitation, we will add more results about the error cases and the application limitations in the revised paper.

---

### Official Review · Reviewer_PqsA · 2023-08-12

**Soundness:** 3

**Excitement:**

3: Ambivalent: It has merits (e.g., it reports state-of-the-art results, the idea is nice), but there are key weaknesses (e.g., it describes incremental work), and it can significantly benefit from another round of revision. However, I won't object to accepting it if my co-reviewers champion it.

**Paper Topic And Main Contributions:**

The authors aim to exploit the knowledge of the pre-trained language model to balance expression diversity and semantic preservation. The key assumption is that the number of abstract transforming patterns of paraphrase generation (governed by prompts) is finite and usually not large, which motivates the paper to introduce vector-quantized prompts as the cues to control the generation of pre-trained models.

**Reasons To Accept:**

1. The topic of the research is important for various tasks in NLP.

**Reasons To Reject:**

1. The assumption of the paper is not well explained.
2. The idea of using prompt learning is not new, which needs more concrete motivations.
3. The terms such as "discrete rule representations by vector quantization", "abstract transforming patterns" should be discussed in more details for better understanding the paper.

**Reproducibility:**

3: Could reproduce the results with some difficulty. The settings of parameters are underspecified or subjectively determined; the training/evaluation data are not widely available.

**Reviewer Confidence:**

2: Willing to defend my evaluation, but it is fairly likely that I missed some details, didn't understand some central points, or can't be sure about the novelty of the work.

---

> ### Author Rebuttal · Authors · 2023-08-29
>
> We sincerely appreciate your insightful feedback. We have addressed your concerns below and hope our responses provide clarity:
>
> **Q1. The assumption of the paper is not well explained.**
>
> Our paper assumes that the number of abstract transforming patterns of paraphrase generation is finite and usually not large. Bhagat [1] summarized all the categories of paraphrases in English and indicated that the high level of the transforming patterns of paraphrasing is limited. Transforming pattern means the change of syntactic structure between two sentences that have the same semantic meaning. Therefore, the above assumption is well established. We have also illustrated the paraphrasing process using different transforming patterns in Figure 1 for an intuitive understanding of the above claim.
>
>
> **Q2. The idea of using prompt learning is not new.**
>
> Admittedly, the idea of prompt learning was actually not first invented by us. But we introduce it into paraphrase generation to produce instance-dependent prompts for the first time. In addition, we combine it with the vector quantizing (VQ) technique, which is no doubt a pretty novel idea. We also find that the traditional VQ does not work in prompt learning. This paper further proposes a new k-means-based approach for training the VQPrompt model which avoids index collapse. Therefore, we believe the proposed method is new and novel.
>
> **Q3. The meaning of  "discrete rule representations by vector quantization" and "abstract transforming patterns".**
>
> The word “discrete” in this paper means that the representations that describe the paraphrasing transforming patterns are not continuous. We use this term to indicate the significant distinction with most of the deep learning frameworks where the deep representations are continuous and obtain gradients flexibly. Those discrete representations are depicted in Figure 3.
>
>
> “The abstract transforming patterns” stands for the patterns that characterize the high-level mapping from a sentence to its paraphrase. Considering that VQPrompt generates paraphrases according to the discrete rule representations, these learned rule representations are the abstract transforming patterns of the paraphrases. We will add the above discussion in the revised paper. Thanks for the reviewer’s helpful comment.
>
> [1] Bhagat R, Hovy E. What is a paraphrase?. Computational Linguistics, 2013, 39(3): 463-472.

---

### Meta-Review · Area_Chair_Ge3v · 2023-09-18

**Recommendation:** 3

**Metareview:**

This work introduces a method for enhancing paraphrase generation using quantized prompts. The method employs a vector-quantizer to encode prompts and generate paraphrases from a codebook of prompt templates that are updated with a k-mean algorithm. However, the novelty of the method and the evidence for the claims about the syntactic information captured by the learned prompts are questionable. Overall, this is important research top in NLP. It also provides comprehensive experiments.

---

### Decision · Program_Chairs · 2023-10-07

**Decision:**

Accept-Findings

**Comment:**

This work introduces a method for enhancing paraphrase generation using quantized prompts. The method employs a vector-quantizer to encode prompts and generate paraphrases from a codebook of prompt templates that are updated with a k-mean algorithm. However, the novelty of the method and the evidence for the claims about the syntactic information captured by the learned prompts are questionable. Overall, this is important research top in NLP. It also provides comprehensive experiments.